# Cross-Cultural Adaptation and Validation of the Portuguese Version of the Psychosocial Impact of Dental Aesthetics Questionnaire

**DOI:** 10.3390/ijerph19169931

**Published:** 2022-08-11

**Authors:** João Fidalgo, João Botelho, Luís Proença, José João Mendes, Vanessa Machado, Ana Sintra Delgado

**Affiliations:** 1Clinical Research Unit (CRU), Centro de Investigação Interdisciplinar Egas Moniz (CiiEM), Egas Moniz Cooperativa de Ensino Superior, 2829-511 Almada, Portugal; 2Evidence-Based Hub, Centro de Investigação Interdisciplinar Egas Moniz (CiiEM), Egas Moniz Cooperativa de Ensino Superior, 2829-511 Almada, Portugal; 3Orthodontic Department, Egas Moniz Dental Clinic (EMDC), Egas Moniz Cooperativa de Ensino Superior, 2829-511 Almada, Portugal

**Keywords:** psychosocial impact, dental esthetics, validation, psychometric properties

## Abstract

The aim of this study was to cross-culturally adapt and validate the psychosocial impact of dental esthetics questionnaire (PIDAQ) to the Portuguese language. The PIDAQ was culturally translated and adapted according to international guidelines. In this cross-sectional study, we enrolled 501 subjects in a population-based epidemiological survey conducted at the Egas Moniz Dental Clinic (Almada, Portugal) in June 2022. The participants answered the Portuguese version of the PIDAQ (PIDAQ-PT), which was a 23-item scale with four conceptual domains (self-confidence, social impact, psychological impact and esthetic concern factor). Psychometric properties were estimated using content validity, construct validity, internal consistency and test–retest reliability. The PIDAQ-PT presented an intraclass correlation coefficient (ICC) of 0.84, and a 95% confidence interval (0.73–0.90, *p* < 0.001), with values for the Cronbach’s alpha coefficient of the subconstructs ranging from 0.93 to 0.98. In the confirmatory factor analysis (CFA), the final models presented a good fit, with the comparative fit indices (CFIs) ranging from 0.905 to 0.921 and the root mean squared error of approximation (RMSEA) ranging between 0.088 and 0.090. The PIDAQ-PT was shown to be a valid and reliable tool to assess oral health values in a Portuguese population. Further studies should evaluate the psychometric properties of the oral personal representation on dental specialties and its impact on dental appointments and procedures.

## 1. Introduction

Health-related quality of life regards information on a patient’s self-perceived welfare in relation to a particular medical illness [1,2]. Currently, the notion that oral diseases strongly affect quality of life is undoubtful. For this reason, interest from the research and clinical community has increased with the development of questionnaires that provide insightful information on patients’ self-perceived quality of life related to oral health [3,4,5,6] and to their self-perceived appearance [5,7].

The psychosocial impact of dental esthetics questionnaire (PIDAQ) was originally developed by a multidisciplinary group of experts (psychologists and orthodontists) to quantify the impact of dental appearance [1] and, ever since, has been validated for a series of languages in many countries [2,8,9,10,11,12,13,14,15,16,17]. This tool is considered a multidimensional instrument because it includes four domains: dental self-confidence, social impact, psychological impact and esthetic concern [1].

Considering the importance the PIDAQ has globally, its adequate factorial stability across adult samples and the lack of a Portuguese validated version, we aimed to cross-culturally adapt and validate the PIDAQ to Portuguese (from Portugal), which was nominated as the PIDAQ-PT. We hypothesized that the PIDAQ would present valid and adequate psychometric properties based on previous validations worldwide.

## 2. Materials and Methods

### 2.1. Design and Participants

The present cross-sectional study target population consisted of subjects attending the Egas Moniz Dental Clinic (EMDC), a university dental clinic located in Almada, Portugal. This study received approval from the Institutional Review Board (Ethics Committee of Egas Moniz, ID: 1050) and was conducted in accordance with the Declaration of Helsinki of 1975, as revised in 2013. As inclusion criteria, we were seeking patients over 18 years old or older, Portuguese speakers and those looking for a triage appointment. Subjects who met the inclusion criteria were invited to participate voluntarily and anonymously. Before proceeding with the study, each participant gave informed consent. The interviewer (J.F.) was blinded to the detailed oral health status.

### 2.2. Cross-Cultural Adaptation of the PIDAQ

The original PIDAQ tool is a psychosocial instrument developed to identify an individual’s perceived impact of dental esthetics on psychological and social constructs. This 23-item tool is framed within 4 subscales: dental self-confidence (items 1 to 6), social impact (items 7 to 14), psychological impact (items 15 to 20) and esthetic concern (items 21 to 23) (Table 1). Each item is rated using a 5-point Likert scale, ranging as follows: 0 = “not at all”; 1 = “a little”; 2 = “somewhat”; 3 = “strongly”; and 4 = “very strongly” [1] (Appendix A). To study the psychometric properties of the PIDAQ and to calculate the total score, all items in the dental self-confidence subscales were reversed to bring the direction of the scores into line with the other 3 subscales [2]. Total scores ranged from 0 to 2 after recording, with higher total scores indicating a greater degree of negative psychosocial impact from dental esthetics.

The PIDAQ was first translated into Portuguese by an expert panel embracing four independent bilingual individuals fluent in Portuguese and English (involving two women and two men; V.M., J.B., A.S.D., J.B.) from various oral health backgrounds (general dental practitioner, orthodontist, periodontologists), with years of experience ranging from 2 to 22 years. The original English questionnaire was translated independently in a “double-blinded” approach to Portuguese by two native speakers in Portuguese and English (V.M. and J.B.) and were integrated into a single translation version. Any disagreements were resolved by discussion. Two independent bilingual experts, blinded to the original version, back-translated the Portuguese versions. The two new English versions of the PIDAQ were analyzed by a panel of experts, who assessed the semantic and conceptual equivalence of the original and Portuguese versions of the PIDAQ (Appendix A).

The Portuguese version of the PIDAQ was pilot tested on a random sample of 50 individuals (10% of the total sample required for validation, detailed in Section 2.4). All subjects fulfilled the inclusion criteria (living in Portugal, native in Portuguese and 18 years old or older) and gave informed consent to participate. For reliability analysis purposes, this sample of participants was invited to answer the same test one week later. The pilot test group of patients did not account for the validation per se. The pilot test demonstrated that the Portuguese version of the PIDAQ exhibited appropriate semantic and conceptual equivalence (see Section 2.5.1).

### 2.3. Sociodemographic Variables

Sociodemographic characteristics comprised age and sex.

### 2.4. Sample Size Calculation

To determine the minimum required number of participants, sample size estimation was based on Terwee et al. [18], to ensure a minimum of 10 individuals per questionnaire item. The total minimum number of subjects (*n* = 500) considered the number of parameters and dimensions present in the questionnaire to allow adequate stability of the variance/covariance matrix when performing confirmatory factor analysis (CFA).

### 2.5. Statistical Analyses

#### 2.5.1. Reliability

PIDAQ-PT reliability analysis was conducted through test–retest reliability and internal consistency analysis using 50 participants (10% of the sample size, Section 2.4) who filled out the PIDAQ-PT in two periods, at baseline and one week later [18]. The validity and reliability assessments were carried out between April and May 2022 at the EMDC. The internal consistency was evaluated by calculating Cronbach’s alpha (α) coefficient in R version 1.1–1 (R Studio Team 2018) “ltm” package. The literature suggests that an α coefficient of 0.70 was acceptable for the items in the PIDAQ-PT [19]. To test–retest the proposed reliability, the intraclass correlation coefficient (ICC) was calculated with the two measurement scores from the participants in R version 0.84.1 (R Studio Team 2018) “irr” package. ICC values were interpreted as follows: excellent (over ≥0.9), acceptable (0.80–0.89), weak (0.6–0.79) and inexistent (below 0.60) [20].

#### 2.5.2. Descriptive Analysis and Construct Validity

Descriptive analyses of the background characteristics of the target participants and the PIDAQ-PT items and subscales were performed and presented as counts and correspondent percentages (%), mean and standard deviation (SD), median and interquartile range (IQR) and minimum and maximum values. For all descriptive analyses, the statistical software used was the R version 1.0.8 (R Studio Team 2018) “dplyr” package. The level of statistical significance was set at 5% in all inferential analyses.

The CFA strategy was based on a recent framework conducted [21]. To conduct the CFA, we employed the “lavaan” package for R version 0.6–10 (R Studio Team 2018), to compute the factorial loads and model fitness of each subconstruct. The maximum likelihood method was applied to calculate the model, and the differences between models were calculated through chi-square (χ^2^) and using a likelihood ratio test. Several model fit indices were used to assess the CFA model fit, including the χ^2^/df ratio (good adjustment with values < 2) [22], the root mean squared error of approximation (RMSEA; a good model adjustment considered for values between 0.05 and 0.10%; 90% confidence interval (CI)) [23], the confirmatory fit index (CFI) (cut-off criterion of ≥0.90 indicates a good fit) [24], goodness-of-fit (GFI) statistics (values of 0.90 or greater indicate well-fitting models) [25] and normed-fit index (NFI) (cut-off criterion of 0.90) [26].

Then, the invariance of the PIDAQ-PT was explored across sexes. We estimated four successive models: (1) unconstrained; (2) factor-loading-constrained (Model 1); (3) factor-loading- and structural-covariance-constrained (Model 2); and (4) factor-loading-, structural-covariance- and measurement-residual-constrained (Model 3). To measure the invariance, we used the CFI delta values (ΔCFI), with a cut-off point less than 0.01, which indicated invariance [24,27]. The chi-square delta values (Δχ^2^) were also used and a value lower than standardized Δχ^2^ for 1 − α = 0.095 indicated invariance between the models [28,29].

We also explored the relationships between the PIDAQ-PT items using Spearman’s rank correlation coefficient (rho).

## 3. Results

### 3.1. Reliability of the PIDAQ

On the test–retest analysis, all 50 individuals completed the PIDAQ-PT in both time periods. Of these 50 participants, 26 (52.0%) were female and 24 (48.0%) were male, with similar age intervals (females: 34.2 ± 17.7 vs. males 33.3 ± 10.5). The median total score of the PIDAQ-PT was 16 (range 0–72).

The overall internal consistency was 0.91 (95% CI: 0.87; 0.94), whereas each subscale presented excellent coefficient levels for all subscales (Table 1). In addition, ICC analyses showed a total result of 0.84 (95% CI: 0.73; 0.90) (*p* < 0.001), while all subscales reported values over 0.85, between 0.87 and 0.96. Nominally, one subscale had excellent reliability (dental self-confidence subscale = 0.96), and the remaining had acceptable reliability (Table 1 and Appendix A).

### 3.2. Participant’s Description

A total of 501 participants completed the PIDAQ-PT, with an average age of 48.7 (±18.0), ranging from 18 to 88, (47.1 ± 17.9 and 50.4 ± 18.1 for females and males, respectively), and the group was predominantly composed of women (52.3%, *n* = 262). When analyzing the results of the PIDAQ-PT, item 2 had the highest average score of 3.5 (±0.7), while item 11 had the lowest score with 1.8 (±0.7) in the “Dental self-confidence” and “Social Impact” subscales, respectively (Table 2).

### 3.3. Construct Validity

#### 3.3.1. Factor Validity

The CFA analysis attested the unifactorial structure of the PIDAQ (Table 3). The first-order unifactorial model resulted in an adequate model fit: GFI = 0.846; CFI = 0.921; RMSEA = 0.082; 90% CI (0.077–0.088); and NFI = 0.900 (Table 4).

#### 3.3.2. Psychometric Analysis

The PIDAQ-PT exhibited an adequate reliability (with a Cronbach’s α coefficient of 0.75) and an adequate psychometric feature.

#### 3.3.3. Gender Invariance Measurement

A multigroup CFA assessed the invariance across gender in the PIDAQ-PT (Table 3). Accordingly, there was invariance for gender groups based on the comparisons of CFIs, χ^2^ and the degrees of freedom across the unconstrained and constrained models studied.

#### 3.3.4. Relationships between the PIDAQ Components

Following this, we assessed the correlation between the items of the PIDAQ-PT through Spearman’s rank correlation coefficient. There was a substantially high number of significant correlations (97.8% of all correlations, 262 out of 268) (Appendix A). The correlation between the subscales was also performed, confirming significant correlations amid all subscales (Table 4).

## 4. Discussion

The results of this study demonstrated that the PIDAQ-PT was successfully cross-culturally adapted and validated, thus, providing adequate psychometric properties regarding participants’ self-perceived impact of dental appearance. Furthermore, we observed adequate internal consistency and reliability. Additionally, at the subscale level, the reliability and validity of the PIDAQ-PT were still maintained.

The successful validation of this tool aligned with the hypothesis initially proposed and with the bulk of literature that also confirmed the validity of such an instrument [2,8,9,10,11,12,13,14,15,16,17].

The advantages of the PIDAQ clearly surpassed its limitations as a self-reporting tool. One the one hand, it snapshots the subjective psychosocial view of the esthetics of the dentition of a patient [13], thus, contributing to a better understanding of the patients’ baseline and ongoing esthetic and social prospects in orthodontics. Yet, in our view, this could be expanded to other dental specialties that deal with esthetics, such as periodontal plastic surgery or adhesive dentistry (particularly in veneers and anterior restorations). On the other hand, this instrument may be employed in epidemiological and public health outputs as previously mentioned [13]. Nevertheless, in the view of Wahab et al. [13], this esthetic concern dimension could have intersections with other indices that are now used to determine treatment needs, such as the dental esthetic index and index of orthodontic treatment need [13], as well to attest the motivation towards treatment throughout the clinical course of treatment.

The PIDAQ presents some shortcomings due to its design, purpose and length. Because it provides subjective self-reported measures, it always implies some level of bias that is influenced by the variation of sociodemographic features and lifestyle habits. Initially developed for a younger adult audience, this tool has already been validated for adolescents and older adults, yet the challenge is to make this tool adequate for all age ranges that are able to provide self-reported perceptions of their own dental appearance.

### Strengths and Limitations

This validation study had important limitations worth discussing. The structure of the PIDAQ had a medium size, and this may have contributed to the decreased response rates [30]; thus, efforts shall be undertaken to acquire short versions of this tool without compromising its reliability. Recently, Alsanabani et al. 2022 [31] tested four short versions of the PIDAQ using six-item and nine-item versions in adolescents, and we foresee that such strategies could be expanded to other age ranges. Due to the face-to-face interview strategy implemented, there could have existed some higher social desirability bias, “yes-saying” bias and interviewer bias [32]. To avoid them, the questionnaire can be administered in privacy. Nevertheless, the face-to-face strategy contributed to an increased participant coverage, lower cognitive burden, increased response rate and increased questionnaire completion [32].

The setting also benefited the purpose of this study, as the EMDC is located in the Southern Lisbon Metropolitan Area, and the relative cultural and linguistic homogeneity may contribute to the generalizability of this questionnaire in Portugal.

## 5. Conclusions

The PIDAQ-PT showed adequate reliability, internal consistency and construct validity. We, thus, confirmed the suitability of the PIDAQ-PT to measure the quality of life of the Portuguese population.

## Figures and Tables

**Table 1 ijerph-19-09931-t001:** Test–retest reliability using ICC for the PIDAQ-PT.

	Cronbach’s α Coefficient (95% CI)	ICC (95% CI)	* *p*-Value
Subscales			
Dental self-confidence	0.98 (0.95; 0.99)	0.96 (0.92; 0.97)	<0.001
Social impact	0.93 (0.87; 0.96)	0.87 (0.78; 0.92)	<0.001
Psychological impact	0.93 (0.89; 0.97)	0.88 (0.80; 0.93)	<0.001
Esthetic concern	0.93 (0.83; 0.97)	0.87 (0.77; 0.92)	<0.001
Total score	0.91 (0.87; 0.94)	0.84 (0.73; 0.90)	<0.001

CI—confidence interval; ICC—intraclass correlation coefficient. * *p* < 0.001.

**Table 2 ijerph-19-09931-t002:** Descriptive statistics of the PIDAQ scores (mean and standard deviation (SD), median and interquartile range (IQR), minimum and maximum).

	Mean (SD)	Median (IQR)	Min-Max
PIDAQ-PT total score	29.3 (7.8)	25 (80)	0–92
Dental self-confidence subscale	11.0 (1.4)	10 (19)	0–24
Item 1	3.3 (0.7)	4 (3)	1–5
Item 2	3.5 (0.7)	4 (3)	1–5
Item 3	3.2 (0.0)	3 (3)	1–5
Item 4	2.9 (0.0)	3 (3)	1–5
Item 5	3.1 (0.0)	3 (3)	1–5
Item 6	3.1 (0.0)	3 (3)	1–5
Social impact subscale	6.7 (3.5)	4 (32)	0–32
Item 7	2.1 (0.7)	1 (4)	1–5
Item 8	1.9 (0.7)	1 (4)	1–5
Item 9	1.7 (0.0)	1 (4)	1–5
Item 10	1.7 (0.7)	1 (4)	1–5
Item 11	1.6 (0.7)	1 (4)	1–5
Item 12	2.0 (0.0)	1 (4)	1–5
Item 13	1.7 (1.4)	1 (4)	1–5
Item 14	1.9 (0.7)	1 (4)	1–5
Psychological impact subscale	8.0 (1.4)	7 (21)	0–24
Item 15	2.6 (2.1)	2 (4)	1–5
Item 16	1.7 (1.4)	1 (4)	1–5
Item 17	2.3 (0.7)	2 (4)	1–5
Item 18	2.3 (0.7)	2 (4)	1–5
Item 19	1.8 (0.0)	1 (4)	1–5
Item 20	3.3 (0.7)	4 (3)	1–5
Esthetic concern subscale	3.5 (1.4)	2 (12)	0–12
Item 21	2.1 (0.7)	1 (4)	1–5
Item 22	2.2 (0.0)	2 (4)	1–5
Item 23	2.2 (0.7)	2 (4)	1–5

SD—standard deviation; IQR—interquartile range; PIDAQ-PT—oral health value scale Portuguese version.

**Table 3 ijerph-19-09931-t003:** Model fit indices in the unifactorial model and configurational invariance by sex.

Description	χ2	df	χ^2^/df	CFI	GFI	RMSEA (90% CI)	NFI	ΔCFI	Δχ^2^	Δdf
Unifactorial model	986.270 *	224	4.40	0.921	0.846	0.082 (0.077–0.088)	0.900	-	-	-
Measurement invariance across sex										
Unconstrained	1360.080 *	448	3.04	0.907	0.948	0.090 (0.085–0.096)	0.868	-	-	-
Model 1	1376.954 *	467	2.95	0.907	0.948	0.088 (0.083–0.094)	0.867	0.000	16.874	19
Model 2	1419.656 *	486	2.92	0.905	0.947	0.088 (0.082–0.093)	0.863	0.002	42.702	19
Model 3	1419.656 *	486	2.92	0.905	0.947	0.088 (0.082–0.093)	0.863	0.000	0	0

CFI, confirmatory fit index; CI, confidence interval; df, degrees of freedom; GFI, goodness of fit index; NFI, normed-fit index; RMSEA, root mean square error of approximation; χ^2^, Chi-square. Model 1, factor-loading-constrained; Model 2, factor-loading- and structural-covariance-constrained, Model 3, factor-loading-, structural-covariance- and measurement-residual-constrained. * *p* < 0.01.

**Table 4 ijerph-19-09931-t004:** Correlation between PIDAQ subscale scores.

PIDAQ Subscale	Professional Dental Care	Appearance and Health	Flossing	Retaining Natural Teeth
Professional Dental Care	1.00	0.42 ***	0.63 ***	0.65 ***
Appearance and Health	-	1.00	0.69 ***	0.61 ***
Flossing	-	-	1.00	0.76 ***
Retaining Natural Teeth	-	-	-	1.00

Values are Spearman’s rank correlation coefficient (rho), *** *p* < 0.001.

## Data Availability

Not applicable.

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
