# Peer review of "Cross-Cultural Adaptation and Validation of the Portuguese Version of the Psychosocial Impact of Dental Aesthetics Questionnaire"

_ijerph, 2022, doi:10.3390/ijerph19169931_

Round 1

Reviewer 1 Report

Validating the Portuguese version of the Psychosocial Impact of Dental Aesthetics Questionnaire (PIDAQ) to assess patients' self-perception of their oral health, so that it could be applicable to regional or local people is a wonderful thought. This is one of the many studies that validated the original PIDAQ questionnaire in English to their local languages. 

1. Introduction: Line 38,39 - rephrasing required for "and to the appearance self-perception [5,7]. 

2. Importance of the need for a validated PIDAQ questionnaire in the Portuguese language needs to be addressed in a sentence or so. 

3. Originally, the PIDAQ questionnaire focused on young adults undergoing orthodontic treatment. Your research study focuses on patients attending the dental clinic who are over 18 years of age. Will they be undergoing any type of dental treatment, consultation etc?

4. It looks like co-authors have published a similar paper using Oral health Values questionnaire.  https://www.ncbi.nlm.nih.gov/pmc/articles/PMC9143491/. Although this paper is cited, it is not cited as needed in certain places. For example descriptive analysis and construct validity. It is similar to the previously published paper. It would be great to add the previously published reference paper here and revise the sentences. 

5. 'Line 97' mentions n=500 and 'line 155' mentions 501 participants. 

6. Table 1: A legend for significance with asterisk, as you have displayed in Table 3 would be good. Asterisk symbol is needed to be added within the table. 

7. A comma after item23 can be removed in table 2

8. What does delta CFI, delta chi-square, delta degrees of freedom indicate? and need to explain how the difference is obtained? 

9.  Subscales could be added in the supplementary file for items 1 to 23

10. OHVS questionnaire should be replaced as PIDAQ or PIDOQ-PT questionnaire in the supplementary file. 

Author Response

Reviewer #1 (Remarks to the Author):
Validating the Portuguese version of the Psychosocial Impact of Dental Aesthetics Questionnaire (PIDAQ) to assess patients' self-perception of their oral health, so that it could be applicable to regional or local people is a wonderful thought. This is one of the many studies that validated the original PIDAQ questionnaire in English to their local languages. 

1. Introduction: Line 38,39 - rephrasing required for "and to the appearance self-perception [5,7]. 
Our answer: We appreciate this remark. We have rephrased lines 38-39 that now reads as follows: “and ​​to the self-perceived appearance [5,7]”.

2. Importance of the need for a validated PIDAQ questionnaire in the Portuguese language needs to be addressed in a sentence or so. 
Our answer: We have provided a brief explanation on the importance of the need for a validated PIDAQ questionnaire in the Portuguese language in lines 46-47 as follows: “Considering the importance PIDAQ is having globally, its adequate factorial sta-bility across adult samples and the lack of a Portuguese validated version”. As we could perceive from your remark, this may account for the requested sentence.

3. Originally, the PIDAQ questionnaire focused on young adults undergoing orthodontic treatment. Your research study focuses on patients attending the dental clinic who are over 18 years of age. Will they be undergoing any type of dental treatment, consultation etc?
Our answer: We appreciate this question. The focused patients attended the dental clinic seeking, originally, a triage appointment. At this first appointment, patients are submitted to a dental triage protocol, with a health questionnaire and oral and dental examinations, to guide their treatment needs. We emphasize the screening purpose of this triage.

4. It looks like co-authors have published a similar paper using Oral health Values questionnaire.  https://www.ncbi.nlm.nih.gov/pmc/articles/PMC9143491/. Although this paper is cited, it is not cited as needed in certain places. For example descriptive analysis and construct validity. It is similar to the previously published paper. It would be great to add the previously published reference paper here and revise the sentences. 
Our answer: We followed your suggestion to cite the above mentioned reference. We added the following sentence: “CFA strategy was based on a recent framework conducted [21].”. Regarding the sentences, we have revised them.

5. 'Line 97' mentions n=500 and 'line 155' mentions 501 participants. 
Our answer: We appreciate this remark, however, in line 97 we mention the minimum required sample estimated, that was 500. Line 155 mentions the actual number of participants enrolled.

6. Table 1: A legend for significance with asterisk, as you have displayed in Table 3 would be good. Asterisk symbol is needed to be added within the table. 
Our answer: We have followed your recommendation.

7. A comma after item23 can be removed in table 2
Our answer: We appreciate this remark and it was resolved accordingly.

8. What does delta CFI, delta chi-square, delta degrees of freedom indicate? and need to explain how the difference is obtained? 
Our answer: We have originally explained why these delta indicate. In lines 130-137 these deltas added information to the invariance of PIDAQ-PT was explored across sex. We appreciate this remark in advance.

9.  Subscales could be added in the supplementary file for items 1 to 23
Our answer: We respectfully understand your suggestion however this information was already explained in lines 64-66. For this reason we have questioned whether adding this information would duplicate information in the article and supplementary files.

10. OHVS questionnaire should be replaced as PIDAQ or PIDOQ-PT questionnaire in the supplementary file. 
Our answer: We have replaced it accordingly.

Reviewer 2 Report

The manuscript is on a relevant topic and is very well presented.

Abstract

The objective/or aim of the study must be presented in the same way throughout the manuscript, therefore consider revising it (abstract, introduction and discussion), also the title.

It is necessary to mention the design of the study (cross-sectional).

Material and methods

The part of the Study design, population and sample needs major revision. Explain how and what the chosen population represents. Did they have a good variation as to the general population? represent different socioeconomic areas?

Page 2, line 57-58 Subjects who met the inclusion criteria were approached to participate voluntarily and anonymously.” which? Please clarify inclusion and exclusion criteria.

Please mention the sample size in this section.

In the abstract the authors mention 501 and in the methodology 500.

Results

I can't find the age range of the study participants.

Author Response

Reviewer #2 (Remarks to the Author):
The manuscript is on a relevant topic and is very well presented.

Abstract
The objective/or aim of the study must be presented in the same way throughout the manuscript, therefore consider revising it (abstract, introduction and discussion), also the title.
Our answer: We have revised the aim in this section to align with the introduction (line 15). and discussion (lines 197-199).

It is necessary to mention the design of the study (cross-sectional).
Our answer: We added the design of the study in lines 17-18 and reads as follows: “In this cross-sectional study”.

Material and methods
The part of the Study design, population and sample needs major revision. Explain how and what the chosen population represents. Did they have a good variation as to the general population? represent different socioeconomic areas?
Our answer: The target populations consiste on a consecutive sample of patients seeking a dental triage to understand if they would require dental treatments. This total enumerative sampling is a  nonprobability sampling and the relatively affordable treatment of the clinic, based on the fact that it is a university clinic, allows to increase the generalizability of the population that is being followed. Furthermore, the generalizability of samples from this clinic has been previously presented (doi: 10.7717/peerj.5258 and doi: 10.1371/journal.pone.0269934).

Page 2, line 57-58 “Subjects who met the inclusion criteria were approached to participate voluntarily and anonymously.” which? Please clarify inclusion and exclusion criteria.
Our answer: We have initially stated the inclusion criteria, yet after careful revision, this may have not been the clearest way. For this reason in lines 57-58 we have rewritten: “As inclusion criteria we were seeking: patients over 18 years old or older, Portuguese speakers, and looking for a triage appointment.”.

Please mention the sample size in this section.
Our answer: We have originally mentioned the sample size in the subsection 2.4 entitled “Sample size calculation”.

In the abstract the authors mention 501 and in the methodology 500.
Our answer: We appreciate this remark. However, in line 97 we mention the minimum required sample estimate, which was 500. Line 155 mentions the final sample number of participants enrolled.

Results
I can't find the age range of the study participants.
Our answer: We added this information into line 160, and reads as follows: “ranged from 18 to 88”.

Reviewer 3 Report

In this article, the psychometric validity and reliability of the Psychosocial Impact of Dental Aesthetics Questionnaire (PIDAQ) questionnaire to the Portuguese language were determined. This article is well organized, and the logic of the article is clear. But the paper still need some improvements before acceptance for publication.

My detailed comments are as follows:

1.     The description of the methods should be a bit more detailed.

2.     The PIDAQ-PT reliability analysis was determined using 50 participates, what about other items in this questionnaire?

3.     Can the presentation of the results be more intuitive and clearer?

4.     The format of references should be uniform.

5.     The article should be checked for English. I recommend a revise the entire manuscript for language correction.

Author Response

Reviewer #3 (Remarks to the Author):
In this article, the psychometric validity and reliability of the Psychosocial Impact of Dental Aesthetics Questionnaire (PIDAQ) questionnaire to the Portuguese language were determined. This article is well organized, and the logic of the article is clear. But the paper still need some improvements before acceptance for publication.

My detailed comments are as follows:
1.     The description of the methods should be a bit more detailed.
Our answer: We have made some adjustments according to the reviewers suggestions. We hope we have resolved this remark properly.

2.     The PIDAQ-PT reliability analysis was determined using 50 participates, what about other items in this questionnaire?
Our answer: The reliability framework followed a previous published strategy and described in lines 102-112. These 50 participants permitted the reliability analysis of all questions.

3.     Can the presentation of the results be more intuitive and clearer?
Our answer: Unfortunately, due to the validation nature of this study, this makes difficult to turn clearer and more intuitive. The way it is presented is similar to a series of studies even cited in this manuscript.

4.     The format of references should be uniform.
Our answer: We have standardized references as per your suggestion. 

5.     The article should be checked for English. I recommend a revise the entire manuscript for language correction.
Our answer: We had this manuscript revised for English.